# Incidental Severe Fatty Degeneration of the Erector Spinae in a Patient with L5–S1 Disc Extrusion Diagnosed with Limb-Girdle Muscular Dystrophy R2 Dysferin-Related

**DOI:** 10.3390/diagnostics10080530

**Published:** 2020-07-29

**Authors:** Du Hwan Kim, Dae-Hyun Jang, Ja-Hyun Jang

**Affiliations:** 1Department of Physical Medicine and Rehabilitation, College of Medicine, Chung-Ang University, 102 Heukseok-ro, Dongjak-gu, Seoul 06973, Korea; 2Department of Rehabilitation Medicine, College of Medicine, The Catholic University of Korea, Seoul 06591, Korea; dhjangmd@naver.com; 3Department of Laboratory Medicine and Genetics, Samsung Medical Center, Sungkyunkwan University School of Medicine, Seoul 06351, Korea; tinky80@naver.com; 4Green Cross Genome, Yongin 16924, Korea

**Keywords:** axial myopathy, dysferlinopathy, limb-girdle muscular dystrophy, lumbar disc

## Abstract

Limb-girdle muscular dystrophy type R2 dysferin-related (LGMD R2 dysferin-related), a phenotype of dysferlinopathy, usually begins with pelvic girdle weakness. A 35-year-old male presented with right leg pain for 2 weeks without a previous history of limb weakness. Magnetic resonance imaging of the lumbar spine showed disc extrusion at L5–S1 and incidental severe fatty degeneration of the lumbar erector spinae. Physical examination demonstrated no definite limb weakness. Serum creatine kinase levels were elevated. Genetic testing using a targeted gene-sequencing panel identified compound heterozygous variants NM_003494.3(*DYSF*) c.[1284+2T>C]; [5303G>A]. Computed tomography revealed fatty degeneration of lower-limb muscles, which was mild in the adductor muscles and severe in the gluteus minimus. Immunohistochemistry staining of the vastus lateralis showed under-expression of dysferlin. This patient was diagnosed with LGMD R2 dysferin-related. Thus, unusual fatty degeneration of the lumbar paraspinalis can be a manifestation of dysferlinopathy.

## 1. Introduction

Pathogenic variants in the dysferlin gene (*DYSF*) lead to various phenotypes, mainly Miyoshi myopathy (MM) and limb-girdle muscular dystrophy type R2 dysferin-related (LGMD R2 dysferin-related) [1,2]. Less common forms of dysferlinopathy include asymptomatic hyperCKemia, distal anterior compartment myopathy, a pseudometabolic form, and a proximo-distal phenotype [3]. MM initially affects the posterior compartment of the distal lower limb, while LGMD R2 dysferin-related begins with proximal lower-limb muscles [4]. Although both MM and LGMD R2 dysferin-related involve axial muscles and eventually the upper limbs, a phenotype with a predominant involvement of axial muscles rather than lower-limb muscles is very rare [5,6]. Therefore, we believed it worthwhile to report a patient with LGMD R2 dysferin-related who presented with severe fatty degeneration of the lumbar paraspinalis, which was caused by compound heterozygous variants NM_003494.3:c.1284+2T>C (rs398123765) and NM_003494.3:c.5303G>A/p.Arg1768Gln (rs148860301) in *DYSF*. Written informed consent was obtained from the patient. This study was approved by the Institutional Review Board of Dongsan Medical Center (number 2019-11-004, approved on 21 November 2019).

## 2. Case Presentation

A 35-year-old male visited our spine clinic presenting with back pain radiating to the right buttock and posterior thigh for 2 weeks. Numeric rating scale for the back pain was 8 out of 10. Table 1 summarizes the demographic and clinical examination data. The patient was not obese (height, 172 cm; weight, 70 kg; body mass index, 23.7) and did not have any specific past medical history or past surgeries. He tended to lean to the left side to alleviate pain and could not bend his trunk because of severe pain. On inspection, there was no atrophy of the lower limbs. Motor strength tests of both his lower limbs at supine position were normal (all 5 of 5 on MRC grade) with regard to hip-flexor, -adductor, -abductor, and -extensor; knee extensor; and ankle-dorsiflexor and -plantarflexor strengths. Sensory tests revealed decreased sensation to light touch and pin prick in the right S1 dermatome. Additionally, deep tendon reflex test showed a decreased ankle jerk on the right side, and range of motion of the lumbar spine was nil due to severe pain. Clinical manifestations and physical examinations suggested right S1 radiculopathy. Magnetic resonance image (MRI) of the lumbar spine demonstrated L5–S1 disc extrusion compressing the S1 root and incidental complete fatty degeneration of the lumbar paraspinalis (Figure 1A,B). Initial laboratory studies revealed elevated levels of creatine kinase (CK) (2015.5 U/L; normal range, 32–294) and liver enzymes (alanine aminotransferase, 92 U/L (normal range, 5–44); aspartate aminotransferase, 62 U/L (normal range, 8–38)). He had no history of liver diseases and was neither on medications nor on herbal remedies. Besides, there was no evidence of family history of muscle diseases.

Further investigations to clarify unexplainable fatty degeneration of the lumbar paraspinalis and abnormal liver function tests were necessary. Autoantibody test as well as investigations of viral infections related the liver, such as HAV, HBV, and HCV infections, were negative. Needle electromyography revealed no abnormal spontaneous activity in the lower limbs or lumbar paraspinalis, but showed myopathic motor unit action potentials in the vastus lateralis and hamstring muscles. A computed tomography (CT) scan of the lower extremities showed mild and severe fatty degeneration of the adductor muscles and gluteus minimus, respectively; however, no definite fatty degeneration of hamstrings or posterior-compartment muscles of the distal lower limbs was observed (Figure 2A–C). Based on the laboratory findings, electrodiagnosis, and imaging studies, the occurrence of myopathy was highly suspected in this patient.

Genetic testing was performed using a targeted gene-sequencing panel for neuromuscular diseases that can assess 293 genes, including 138 genes associated with myopathy. Genomic DNA was extracted from the peripheral blood of the patient. Library preparation and target enrichment were performed by the hybridization capture method. Custom oligo design and synthesis were performed by Celemics (Seoul, Korea). Massively parallel sequencing was performed using 2 × 150 bp in the paired end mode of the MiSeq platform (Illumina, San Diego, CA, USA). Sequence reads were aligned with Burrow–Wheeler Aligner (version 0.7.12, MEM algorithm). Duplicated reads were removed with Picard (http://broadinstitute.github.io/picard), and local realignment and recalibration were performed with the Genome Analysis Tool Kit (GATK, version 3.5). Variant calling was conducted with GATK. Variants were annotated by Variant Effect Predictor and dbNSFP. Common variants with minor allele frequency ≥1% were filtered out using public databases (1000 Genomes Project, Exome Variant Server, Exome Aggregation Consortium). Average coverage depth was 100×, and 99% of target bases were covered by 10× sequence reads.

We found compound heterozygous variants in the patient’s *DYSF* gene: a splicing variant in exon 13 (NM_003494.3: c.1284+2T>C) from the mother (previously reported as a pathogenic variant [7]) and 0.0077% frequency (ESP6500_ALL) reported a missense variant in exon 47 (NM_003494.3: c.5303G>A/p.Arg1768Gln) from the father (Figure 3). The NM_003494.3:c.5303G>A variant was previously reported as a variant of uncertain significance (ClinVar: RCV000266653.2 and RCV000725114.1) and the alternative variant (NM_003494.3:c.5302C>T/p.Arg1768Trp) was pathogenic by ClinVar. This variant was predicted to be deleterious by several in silico analysis tools (SIFT damaging (0.001), PolyPhen-2 deleterious (0.999), and MutationTaster Disease causing (0.999)) and the nucleotide is relatively conserved (GERP++_RS 5.1399 and phyloP30wat_mammalian 1.1759). The patient’s parents were determined to be heterozygous carriers of each variant, which occurred in *trans*. One of the two unaffected sisters of the patient was a carrier of c.5303G>A (Figure 3). Each variant was confirmed by conventional Sanger sequencing. Biopsy was performed on the patient’s vastus lateralis muscle. Immunohistochemistry showed a decreased expression of dysferlin (Figure 4A–C). Based on the imaging studies, electrophysiologic assessments, and immunohistochemical findings, we diagnosed this patient as a LGMD R2 dysferin-related phenotype of dysferlinopathy. The evidence presented above and the segregation of the variants suggest that the missense variant c.5303G>A/p.Arg1768Gln is a pathogenic variant, on the basis of the American College of Medical Genetics and Genomics guidelines regarding the interpretation of sequence variants (PS3+PM2+PM3+PM5+ PP3) [8].

The patient received conservative treatment, including physiotherapy and transforaminal epidural steroid injection for the L5–S1 extruded disc. The back pain and radiating pain in the buttock and posterior thigh disappeared over a period of 3 months. Electrocardiogram did not show arrhythmias. At the same time, the radicular pain disappeared, and motor examination also did not show any weakness of lower limbs. We could not find any weakness of lower limbs even on functional tasks, such as sit to stand, heel gait, and single heel rise. The lordotic curvature of his lumbar spine was mildly increased, which had not been noticed by the patient. During the follow-up for 2 years, the patient did not complain of radiating pain or weakness of lower limbs, but continued to complain of diffuse back pain, which was aggravated after heavy work. Follow-up MRI at 2 years after the initial visit revealed a marked resolution of the herniated disc and progression of fatty degeneration of lumbar paraspinalis (Figure 5A,B).

## 3. Discussion

We report a patient with a LGMD R2 dysferin-related phenotype of dysferlinopathy who is unique with regard to two aspects. First, he presented with incidental severe fatty replacement of his lumbar paraspinalis without limb weakness, which is a rare finding. Second, a variant (NM_003494.3: c.5303G>A) which was previously reported as a variant of uncertain significance was confirmed as a pathogenic variant by clinical findings and functional analysis of the variant.

The patient underwent lumbar-spine MRI because of typical S1 radicular pain without lower-limb weakness caused by disc extrusion. Unexpected complete fatty degeneration of the lumbar paraspinalis and laboratory tests were strongly suggestive of a muscle disorder. Fatty replacement on lumbar-spine MRI in adults is a common finding mainly related to lumbar degenerative kyphosis, spinal stenosis, aging, or camptocormia. Our patient was too young for the fatty degeneration of his lumbar paraspinalis to be explained by any of the degenerative causes described above. Further evaluation is warranted for incidental disproportionate fatty degeneration of the lumbar paraspinalis as the patient ages, spinal canal narrowing, or lumbar sagittal imbalance.

Although dysferlinopathy has various clinical manifestations and disease courses, LGMD R2 dysferin-related usually begins with pelvic girdle weakness, while MM predominantly affects the posterior compartment of a lower leg. In 2008, Seror et al. reported a patient who presented with chronic lower back pain, and the condition was diagnosed as dysferlinopathy with complete degeneration of the lumbar paraspinalis; axial involvement was thought to be extremely rare [6]. However, a recent MRI study revealed that axial myopathy in dysferlinopathy is not uncommon [1,5]. It is still inconclusive whether axial muscles and lower-limb muscles are primarily affected or which musculature is involved in the early stage of dysferlinopathy. Although the patient in our study had no weakness of the lower limbs, needle electromyography revealed pathologic manifestations of the lower-limb muscles. This finding suggests a possibility of the patient developing weakness in the lower limbs in the future. Imaging study revealed severe fatty degeneration of the lumbar paraspinalis, as well as the gluteus minimus; however, the involvement of hamstrings or the posterior-compartment muscles of the distal lower limb, which are known to be involved at the early stage of dysferlinopathy, was not obvious. Although he initially exhibited incidental severe fatty replacement of the lumbar paraspinalis without limb weakness, it is important to continue to monitor whether he develops weakness in the lower limbs in the future.

Although there are a few studies on the prognosis of symptomatic dysferlinopathy, the prognosis of pauci-symptomatic patients with dysferlinopathy is not well understood [1,3,5]. Moreover, clinical spectrums can be significantly diverse even between individuals in a family with the same pathogenic variant [9]. It is difficult to predict the prognosis of an atypical patient with a missense variant associated with an unusual phenotypic presentation of dysferinopathy. However, a well-defined cohort study of symptomatic patients showed that dysferlinopathy has a poor functional prognosis, as more than half of the patients needed mobility devices, such as a walker or wheelchair; however, no patient displayed a reduced forced vital capacity or electrocardiographic abnormalities [3]. A retrospective cohort study suggested that exercise may actually increase the rate of disease progression in dysferlinopathy [10]. Angelini et al. reported that sportive activity seems to accelerate the severity of disease [11]. However, the effect of exercise in dysferinopathy is not yet established as detrimental [10]. Although the patient in our study has not exhibited weakness of the lower limbs till date, it is important to monitor clinical manifestations in patients while preventing strenuous activities.

As a limitation of this case report, we did not perform a whole-body MRI at initial visit or follow-up CT scans. Whole-body MRI has advantages over CT in that it detects morphologic changes in muscles of the whole body in patients with myopathy [5]. A follow-up CT or MRI can evaluate the progression of disease in the lower extremity. Next, we did not study the biopsied muscle by Western blot. Because dysferin can be reduced in other myopathies, absence of dysferin on the Western blot is the golden standard for dysferinopathy [12]. In addition, we did not quantitatively assess the extent of inflammatory response and muscle fiber degeneration and regeneration, although these results can evaluate the development and progression of the disease [13]. The patient did not undergo a 24-h Holter monitoring and echocardiography to rule out arrhythmias.

In the present case, a CT scan of the lower extremities revealed fatty degeneration only in the gluteus minimus and adductor muscles, which explained the lack of definite weakness in the lower extremities and lack of difficulty in performing daily activities. Previous reports demonstrated that dysferlinopathy can present with atypical forms, including exercise intolerance, fatigue, and hyperCKemia [14]. Considering that this patient was incidentally diagnosed at a spine clinic and fatigue or exercise intolerance of the lumbar paraspinalis in hidden myopathy can lead to lower back pain, a large proportion of patients with dysferlinopathy may prefer to seek treatment at a spine clinic. Therefore, spine physicians should pay attention to unusual fatty degeneration of the lumbar paraspinalis, which can be a manifestation of dysferlinopathy.

## 4. Conclusions

In the current study, we described a case with incidental severe fatty degeneration of the erector spinae who presented with L5–S1 disc extrusion and eventually was diagnosed with LGMD R2 dysferin-related. Therefore, physicians should pay attention to unusual fatty degeneration of the lumbar paraspinalis, which can be a manifestation of dysferlinopathy.

## Figures and Tables

**Figure 1 diagnostics-10-00530-f001:**
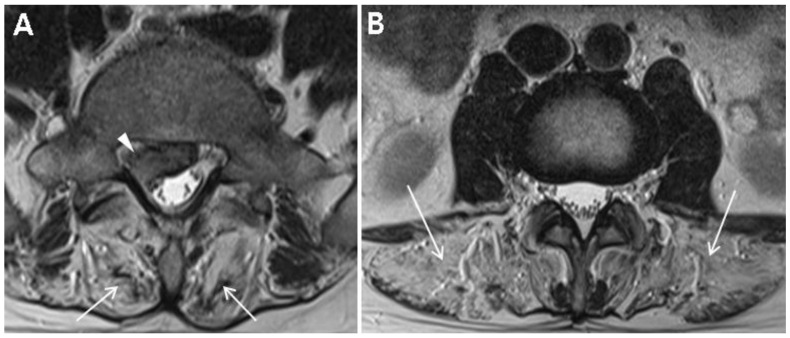
Magnetic resonance image (MRI) of the lumbar spine. (**A**) L5–S1 disc extrusion on the right (arrowhead) and severe fatty degeneration of the erector spinae and multifidus (arrows) on an axial T2WI of lumbar spine MRI. (**B**) Severe fatty degeneration of the lumbar paraspinalis (arrows) at the L3 vertebral body level.

**Figure 2 diagnostics-10-00530-f002:**
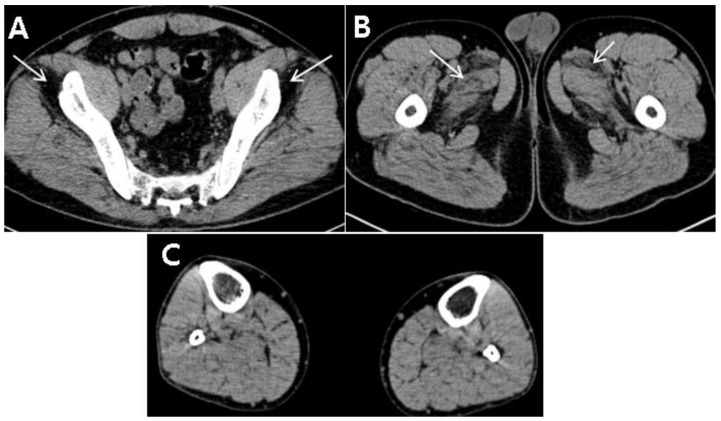
A computed tomography (CT) scan of the lower extremities. (**A**) Severe fatty degeneration of the gluteus minimus (arrows) on muscle computed tomography (CT). (**B**) Mild fatty degeneration of the adductor longus and brevis (arrows) on CT. (**C**) No definite fatty degeneration of the posterior compartment of the distal lower limb on CT.

**Figure 3 diagnostics-10-00530-f003:**
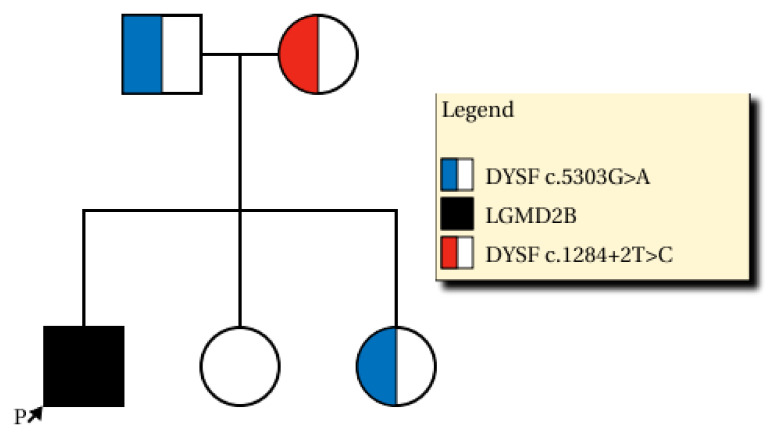
Family pedigree diagnosed with compound heterozygous *DYSF* (dysferlin gene) variants. The black filled-in pedigree member is the patient (NM_003493.3:c.5303G>A and NM_003493.3:c.1284+2T>C), the blue half-filled pedigree member indicates the heterozygous carrier with a pathogenic variant (c.5303G>A), whereas the red half-filled pedigree member is the heterozygous carrier with a previously reported pathogenic variant (c.1284+2T>C).

**Figure 4 diagnostics-10-00530-f004:**
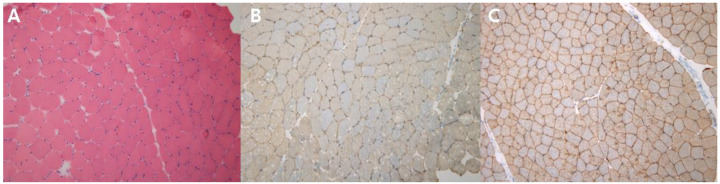
The results of muscle biopsy. (**A**) Mild inflammatory cell infiltration on H&E stain (×100). (**B**) Under-expression of dysferlin, as demonstrated by immunohistochemistry (×100). (**C**) Normal expression of dysferlin (reference) (×100).

**Figure 5 diagnostics-10-00530-f005:**
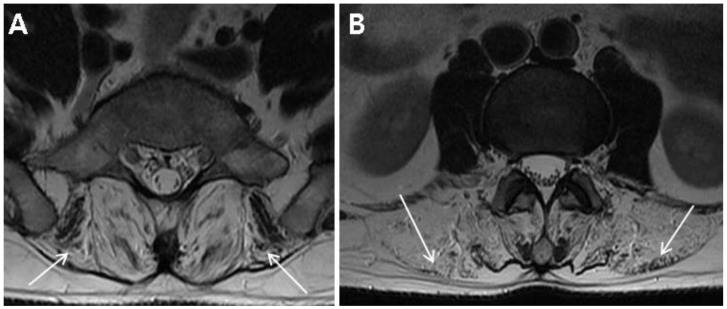
Follow-up magnetic resonance image (MRI) of the lumbar spine. (**A**) At 2 years after the onset of S1 radiculopathy, near-complete resolution of L5-S1 extruded disc and progression of fatty degeneration of the erector spinae (arrows). (**B**) Fatty degeneration of the erector spinae (arrows) at the L3 vertebral body level was also aggravated.

**Table 1 diagnostics-10-00530-t001:** Summary of demographics and clinical examination data.

Demographics		Clinical Examinations	
Age, yrs	35	Motor strength ^a^	
Sex	Male	hip-flexor	5/5
Height (cm)	172	hip-adductor	5/5
Weight (kg)	70	hip-abductor	5/5
Body mass index (kg/m^2^)	23.7	hip-extensor	5/5
Past medical history	None	knee extensor	5/5
Family history of myopathy	None	ankle-dorsiflexor	5/5
Medication or herbal use	None	big toe extensor	5/5
		ankle-plantarflexor	5/5
		Functional tasks ^b^	
		Sit to stand	No Gower’s sign
		Heel gait	No weakness
		Single leg heel rise	No weakness
		Sensory	
		Pin prick	Decreased sensation at S1 dermatome
		Light touch	Decreased sensation at S1 dermatome
		Deep tendon reflexes	
		Knee jerk	2/2
		Ankle jerk	0/2
		Provocation tests	
		FNST	Negative
		SLRT	Positive

^a^ Motor strength of lower limbs presented as MRC grade. ^b^ Evaluation of functional tasks was performed at the stage at which his acute back pain and radicular pain nearly disappeared, and other examinations were performed at the initial visit. FNST, femoral nerve stretching test; SLR, straight-leg raise test.

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
