# Peer review of "Incidental Severe Fatty Degeneration of the Erector Spinae in a Patient with L5–S1 Disc Extrusion Diagnosed with Limb-Girdle Muscular Dystrophy R2 Dysferin-Related"

_diagnostics, 2020, doi:10.3390/diagnostics10080530_

Round 1

Reviewer 1 Report

In this paper, Du Hwan Kim and coworkers describe a LGMD R2 Dysferlin-related (formerly LGMD 2B) affected male with an unusual clinical presentation involving severe fatty degeneration of the the lumbar/axial musculature in absence of clinical evidence of proximal or distal weakness.  Identification of a compound heterozygous DYSF geneotype supporting autosomal recessive inheritance, which were clearly confirmated by familial segregation analysis. This report contributes to expand the highly variable phenotypic spectrum noted for dysferlinopathies.  

The report is very nice documented at clinical, laboratorial, histopathological, genetic and imagenological levels. Also, the manuscript is well writted and structured.

Only I´m consider that the authors, if they consider, could be take into account the following minor points to improve their report:   

  • Please change the word "mutation" by pathogenic variant, allele, change, single-nucleotide variant, etc.
  • Please attend the new ENMC proposed classification (at least, in the first mention of LGMD2B):
    LGMD 2B is now referred as "LGMD R2 Dysferlin-related", please see
    Straub V, Murphy A, Udd B. creat229 th ENMC international workshop: Limb girdle muscular dystrophies – nomenclature and reformed classification, 17-19 March 2017, Naarden, The Netherlands. Neuromusc Disord. 2018;17–19.
  • Please change the concept "Heterozygous mutation" by "compound heterozygous DYSF genotype"
  • Please refer the identified DYSF genotype accordingly to HGVS nomenclature: NM_003494.3(DYSF)c.[1284+2T>C];[5303G>A] or p.[?];[Arg1768Gln]. It is of utmost importance to include it into abstract for indexing purpouses.
  • Due to multiple isoforms described in DYSF gene, please include the employed reference sequence, and the ID accessions (at least, at their first mentions) in databases; i.e.  NM_003494.3(DYSF):c.1284+2T>C (rs398123765)
  • Actually, the variant NM_003494.3(DYSF):c.5303G>A o p.(Arg1768Gln) is now enlisted in the following databases: gnomAD (https://gnomad.broadinstitute.org/variant/2-71894608-G-A), ClinVar (RCV000757894.1: Likely pathogenic* - Autosomal recessive limb-girdle muscular dystrophy type 2B, RCV000266653.2, RCV000725114.1) y dbSNP (rs148860301). Then, the authors must be consider to change the concept of "novel mutation" to "a missense variant associated to an unusual phenotypic presentation of dysferlinopathy". Please note that this concept could be change at the title of their manuscript. 
  • Could be desirable to indicate the pathogenic ACMG score obtained for p.(Arg1768Gln).

Author Response

Thanks for your valuable suggestions and questions. We hope that your suggestions and questions may enhance the quality of our work and our revision or detailed explanations may satisfy you. Please again review our manuscript thoroughly.

  • Please change the word "mutation" by pathogenic variant, allele, change, single-nucleotide variant, etc.
    • Response: Thanks for your comment. We changed the word “mutation” by pathogenic variant through the manuscript.
  • Please attend the new ENMC proposed classification (at least, in the first mention of LGMD2B):
    LGMD 2B is now referred as "LGMD R2 Dysferlin-related", please see
    Straub V, Murphy A, Udd B. creat229 th ENMC international workshop: Limb girdle muscular dystrophies – nomenclature and reformed classification, 17-19 March 2017, Naarden, The Netherlands. Neuromusc Disord. 2018;17–19.
    • Response: Thanks for your comment. We applied this nomenclature and referenced it.
  • Please change the concept "Heterozygous mutation" by "compound heterozygous DYSF genotype"
    • Response: Thanks for your comment. We changed the concept "Heterozygous mutation" by "compound heterozygous DYSF genotype"
  • Please refer the identified DYSFgenotype accordingly to HGVS nomenclature: NM_003494.3(DYSF)c.[1284+2T>C];[5303G>A] or p.[?];[Arg1768Gln]. It is of utmost importance to include it into abstract for indexing purposes.
    • Response: Thanks for your comment. In abstract, we modified sentence as following: “Genetic testing using a targeted gene-sequencing panel identified compound heterozygous variants NM_003494.3(DYSF) c.[1284+2T>C];[5303G>A].”
  • Due to multiple isoforms described in DYSFgene, please include the employed reference sequence, and the ID accessions (at least, at their first mentions) in databases; i.e.  NM_003494.3(DYSF):c.1284+2T>C (rs398123765)
    • Response: Thanks for your comment. We referenced it as following: 3:c.1284+2T>C (rs398123765) and NM_003494.3:c.5303G>A/p.Arg1768Gln (rs148860301).
  • Actually, the variant NM_003494.3(DYSF):c.5303G>A o p.(Arg1768Gln) is now enlisted in the following databases: gnomAD (https://gnomad.broadinstitute.org/variant/2-71894608-G-A), ClinVar (RCV000757894.1: Likely pathogenic* - Autosomal recessive limb-girdle muscular dystrophy type 2B, RCV000266653.2, RCV000725114.1) y dbSNP (rs148860301). Then, the authors must be consider to change the concept of "novel mutation" to "a missense variant associated to an unusual phenotypic presentation of dysferlinopathy". Please note that this concept could be change at the title of their manuscript. 
    • Response: Thanks for your comment. In fact, ClinVar (RCV000757894.1) was registered by one of our authors before the result of muscle biopsy. Thus, this variant was registered as a likely pathogenic at that time. And we know that the other two were registered as a variant of uncertain significance. This study confirmed this variant as a pathogenic on basis of the American College of Medical Genetics and Genomics guidelines regarding the interpretation of sequence variants. This concept applied to this manuscript.
  • Could be desirable to indicate the pathogenic ACMG score obtained for p.(Arg1768Gln).
    • Response: Thanks for your comments. The missense variant c.5303G>A/p.Arg1768Gln is a pathogenic variant, on basis of the American College of Medical Genetics and Genomics guidelines regarding the interpretation of sequence variants (PS3+PM2+PM3+PM5+ PP3).

Reviewer 2 Report

This case report focuses on the fact that paraspinal muscles were found to be abnormal in a case of dysferlinopathy with Rleg pain due to  a radiculopathy for an L5-S1 extruded disc.Posterior leg muscles CT was unrevealing but in proximal LE muscle gluteus was abnormal slightly abnormal semitendinosus in Fig.1that shows some imaging changes.Present CT imaging should be compared to what demonstrated in legs of dysferlinopathy cases by CT scan since MRI imaging was not done.Usually STIR sequences are very informative for dysfelinopathy

The quadriceps biopsy shows some decreased IHC dysferlin stain and inflammatory changes,was western blotting performed?According to Caciottolo et al absence of dysfelin on WB is the  golden standard for dysferlinopathy.The present case is atypical but might be an end spectrum axial myopathy due to dysfelinopathy,Authors should stain biopsy for macrophages and regenerating fibers since according to Fanin (2002) inflammatory features are found in 30% biopsies and have other revealing changes.

The effect of exercise in dysferlinopathy is not yet established as detrimental although sportive activity seems to accelerate the severity of disease (Angelini et al.2011)

Author should use the recent teminolgy for dysferlinopathy that is LGMD R2.

Author Response

Thanks for your valuable suggestions and questions. We hope that your suggestions and questions may enhance the quality of our work and our revision or detailed explanations may satisfy you. Please again review our manuscript thoroughly.

  • This case report focuses on the fact that paraspinal muscles were found to be abnormal in a case of dysferlinopathy with Rleg pain due to  a radiculopathy for an L5-S1 extruded disc. Posterior leg muscles CT was unrevealing but in proximal LE muscle gluteus was abnormal slightly abnormal semitendinosus in Fig.1that shows some imaging changes. Present CT imaging should be compared to what demonstrated in legs of dysferlinopathy cases by CT scan since MRI imaging was not done. Usually STIR sequences are very informative for dysfelinopathy
    • Response: We totally agree your opinion. We did not perform follow-up CT or MRI of lower extremities. He had residual back discomfort, but did not complain of leg weaknss. Thus he refused CT follow-up or MRI of lower extremities. As described in the part of discussion, absence of lower leg MRI and follow-up imaging study is a clear limitation of our study.
  • The quadriceps biopsy shows some decreased IHC dysferlin stain and inflammatory changes,was western blotting performed?According to Caciottolo et al absence of dysfelin on WB is the  golden standard for dysferlinopathy.The present case is atypical but might be an end spectrum axial myopathy due to dysfelinopathy,Authors should stain biopsy for macrophages and regenerating fibers since according to Fanin (2002) inflammatory features are found in 30% biopsies and have other revealing changes.
    • Response: We totally agree your opinion. As you commented, dysferin can be secondly reduced in other myopathies. Thus, western blot is the golden standard for dysferinopathy according to Caciottolo et al. Unfortunately, we did not perform this test. Considering the results of IHC dysferin and genetic analysis together, we could think of it as a pathogenic variant. And we also did not perform other special stains. We reviewed the meaning of stain for macrophages and regenerating fibers. This incomplete pathology analysis was further described as a limitation. Please understand the limitations of our research once more.
  • The effect of exercise in dysferlinopathy is not yet established as detrimental although sportive activity seems to accelerate the severity of disease (Angelini et al.2011)
    • Response: Thanks for your comment. We agree your opinion. Although the role of exercise in dysferinopathy seems to be limited, the exact effect is not yet established because of lack of published study or inappropriate study design. Exert opinions have suggested that strenuous activity can cause the early progression of disease. Considering these concepts, the nuances of the sentences were slightly modified and we referenced the Angelini article.
  • Author should use the recent teminolgy for dysferlinopathy that is LGMD R2.
    • Response: Thanks for your comment. We applied this nomenclature and referenced it.